# The Influencing Factors of Comprehensive Sexuality Education Capacity and Its Correlation with Subjective Social Status among Chinese Junior High School Students

**DOI:** 10.3390/children9081201

**Published:** 2022-08-10

**Authors:** Fan Zhu, Guiyin Zhu, Bibo Jia, Pei Wang, Tianjie Zhao, Yinghua Ma, Bin Dong

**Affiliations:** School of Public Health, Peking University, Beijing 100191, China

**Keywords:** comprehensive sexuality education, sex education, subjective social status

## Abstract

(1) Objective: Evidence suggests that comprehensive sexuality education (CSE) can protect and empower younger generations to advocate for their reproductive health and wellbeing. This survey aims to investigate the current status and influencing factors of CSE among Chinese junior high school students, and to evaluate its correlation with the learning experience of sex education and subjective social status (SSS) to provide evidence for the implementation of CSE in the future. (2) Methods: A total of 4109 participants aged 11 to 16 years were recruited using data from a cross-sectional survey among junior high school students in China in 2021. CSE knowledge, attitude, and skills were used to generate the CSE comprehensive capacity by a principal component analysis. One-way ANOVA was used to assess the different effects of school sex education and family sex education. Multiple linear regression was used to assess the association between CSE comprehensive capacity and SSS. (3) Results: The average score of CSE comprehensive capacity was 82.44 ± 8.60 (with a total score of 100 points) among participants. After the adjustment, subjective social status was positively related to CSE comprehensive capacity (B = 0.28, 95% CI: 0.20–0.36), and SSS (School) (beta = 0.62) had a higher impact on CSE comprehensive capacity compared to SSS (Family) (beta = −0.10). School sex education was associated with the CSE knowledge level with a larger magnitude compared to family sex education (mean deviation = −0.53, *p* = 0.031), whereas family sex education was related to the CSE skill level with a greater magnitude (mean deviation =1.14, *p* = 0.005). (4) Conclusions: These findings suggest that sex education at school and within the family might have a different impact on CSE capacity, which was positively associated with SSS among junior high school students.

## 1. Introduction

China has one of the largest groups of adolescents aged under 19 (around 326 million) worldwide [1]. Evidence from the national surveillance of infectious diseases among Chinese students suggested that sexually transmitted diseases (STDs) increased significantly, and HIV (human immunodeficiency virus) and AIDS (acquired immunodeficiency syndrome) has been the leading cause of death from infectious diseases since 2014 [2]. However, due to Chinese social culture, sex education has been taboo for decades, and sex education is still not openly discussed. However, the Chinese government started to promote health education on sexual and reproductive health (SRH) and a series of policies were issued. In 2020, China revised the Law of the People’s Republic of China on the Protection of Minors [3] to regulate that all schools in China need to conduct sex education for school-aged adolescents. Additionally, for the very first time, the laws in China regulate sex education as compulsory education for school-aged adolescents.

In 2018, UNESCO updated the version of the International Technical Guidance on Sexuality Education (ITGSE), providing evidence and guidance on conducting comprehensive sexuality education (CSE) [4]. Evidence suggests that CSE enables adolescents and young people to develop accurate and age-appropriate knowledge, attitudes, skills, and positive values, such as respect for human rights, gender equality, and diversity, and attitudes and skills that contribute to safe, healthy, and positive relationships [5].

School is considered the ideal place to promote the health and wellbeing of adolescents by conducting health education, health surveillance, and providing health services [6,7]. Furthermore, evidence suggests that sex education within the family can not only promote SRH, but also the relationship among family members [8,9]. Considering that sex education is highly related to social norms and cultural diversity, and the fact of the current heavy workload for school teachers during the COVID-19 pandemic, the family could also be a satisfactory place for sex education.

However, there is still no sufficient evidence to reveal the correlation between family sex education or school sex education and the current status of comprehensive sexuality education on the promotion of health among junior high school students. Hence, to reveal the current status of sex education in China and to provide evidence for conducting CSE in China, we conducted a cross-sectional study on CSE and its influencing factors among junior high school students from six regions in China.

## 2. Methods

### 2.1. Study Design and Participants

Data were collected from a cross-sectional study conducted among Chinese students in junior high schools in six provinces (Beijing, Henan, Liaoning, Chongqing, Sichuan, and Yunnan) in 2021.

Because a school-based sex education survey might cause controversy between the school and parents and lead to a high rejection rate, convenience sampling was used in this survey by inviting all the students within the school to participate with the help of school managers or by contacting persons from the local education or health sectors. Additionally, since non-probabilistic sampling was used and there was a lack of conduction rate regarding sex education among Chinese junior high school students, the sample size was not accurately calculated. All the students and their parents were free to reject to participate, and if they decided to join this investigation, their guardian gave written informed consent.

Eventually, total data from 5577 adolescents aged between 11 and 16 years from 16 schools were collected; 1468 participants who did not answer certain questions as ordered (e.g., “please answer “yes” for this question”) were deemed as invalid data and were excluded. Thus, our sample size for the analysis was 4109 (73.68%).

The Medical Ethical Committee of Peking University Health Science Center approved the study (IRB00001052-21107), and all the adolescents and their parents gave informed consent.

### 2.2. Questionnaire Survey

An online questionnaire was used as an investigation tool to obtain self-reported information on sex education. The questionnaire contained 6 parts: (1) characteristics, including gender, region, age, family socioeconomic (SES) and subjective social status (SSS), parental accompaniment, etc.; (2) themes and the implementation rate of family/school-based sex education; (3) CSE knowledge, 17 items (CSE K); (4) CSE attitude, 11 items (CSE A); (5) CSE skills, 10 items (CSE S); (6) needs for CSE learning.

The items of the CSE knowledge part, CSE attitude part, and CSE skills part were designed to refer to the given “key idea (knowledge, attitudinal and skill)” under the frame of the “8 key concept, learning objectives” in the ITGSE for the ages from 12 to 15. Additionally, all items from the 3 parts were evaluated by experts from the field of children and adolescent health and health education to ensure the content validity; the Kaiser–Meyer–Olkin (KMO) values were 0.86, 0.84, and 0.86, respectively. The internal consistency of Cronbach’s alpha of the three parts was 0.71, 0.80, and 0.80, respectively.

### 2.3. Measurements and Classification

The SES scale was used in the questionnaire with 3 indicators: family income, parents’ occupation, and parents’ educational background. All those indicators were collected according to this procedure [10]: (1) first, calculate the Zscore-SES = Zscore-income × 0.265 + Zscore-education × 0.491 + Zscore-occupation × 0.496; (2) then, use the formula of SES = 50 + 10 × Zscore-SES to obtain the participants’ SES. Subjective social status (SSS) accounts for people’s perception of their family or their standing within a social context [11]; it is also widely used to acquire the objective socioeconomic status (SES), such as income and education of a marginal group or those whose SES is not easily measured, such as children and youth [12]. Compared to objective indicators, SSS may predict aspects of psychological and physical health beyond objective SES indicators, such as family income and parental education. Lower SSS has been linked with greater psychological and physiological stress [13,14], and the chronic toll of low SSS, or the feeling of a low status relative to others, may promote greater threat sensitivity, and ultimately, worse health outcomes regardless of income or education [15]. Participants’ SSS was referred to using the MacArthur SSS Scale–Youth [16]. The SSS scale has a two-item instrument that measures how a young person perceives their family’s (SSS Family) and their own social (SSS School) standing, with each item score ranging from 1–10; a higher score means a higher rank of their family’s or their standing. The total score was calculated as SSS = SSS Family + SSS School (score ranging from 2–20). Parental accompaniment was also measured; if the participants reported that they barely could see their parents almost every day for all kinds of reasons, such as being an orphan, left-behind children, etc., were deemed to have an absence of parental accompaniment. Those who receive accompaniment from one of their parents were deemed to have parental accompaniment in this survey.

### 2.4. Statistical Analysis

The data were exported from the online questionnaire platform in an Excel format and analyzed after data cleaning using Stata/SE 15 (StataCorp, College Station, TX, USA). The continuous variable data were described by ($\;\bar{x}\;\;\pm \;s$), and the categorical variable was described by count and percentage.

The measurement of the CSE knowledge level was based on the total score of the CSE knowledge part, which had 17 items with 1 point for the right answer of each and a full score of 17 points. The CSE attitude level was measured based on the total score of the CSE attitude part, which had 11 items with 5 points each (1-strongly disagree, 2-disagree, 3-unclear, 4-agree, 5-strongly agree) and a full score of 55 points. The CSE skills level was measured based on the total score of CSE skills, which had 10 items with 5 points each (1-very incapable, 2-incapable, 3-not clear, 4-capable, 5-very capable) and a full score of 50 points. The CSE knowledge, attitude, and skill level were all analyzed as continuous variables.

Given that of the CSE knowledge, attitude, and skill parts, all 3 indicators showed good reliability as well as construct validity, principal component analysis was used to calculate the comprehensive capacity of CSE using the 3 indicators. The selection criterion of the principal component was that it could explain the maximum variation of the three indicators (more than 80%). After the calculation, the CSE comprehensive capacity = 1.14 × CSE knowledge + 1.05 × CSE attitude + 0.28 × CSE skills, and the total score was 100.

One-way ANOVA and multiple linear regression models were used to analyze the influencing factor of participants’ CSE comprehensive capacity. The Bonferroni test was used to account for multiple comparisons between sex education and the CSE score.

According to the result of the analysis of influencing factors of participants’ CSE comprehensive capacity, covariates, including survey regions, age, gender, sex education learning experience, and parental accompaniment, were further added to the models (adjusted model).

Multiple linear regression was used to evaluate the correlation between SSS and CSE comprehensive capacity. Models were first adjusted for gender, age, and survey region, and then further adjusted for sex education learning experience and parental accompaniment. Coefficients (β) and 95% CIs were estimated. The test level was α = 0.05, and all *p* values had two-sided probability.

## 3. Results

### 3.1. The Current Status of CSE and Its Influencing Factors

The influencing factors of CSE comprehensive capacity in Chinese junior high school students are shown in Table 1. Among 4109 junior high school students, 28.23% of them self-reported that they had an absence of parental accompaniment; meanwhile, their CSE comprehensive capacity was significantly lower than that of those who have parental accompaniment: 18.50% of them reported that they have never received any sex education, 22.46% of them only received school sex education, 7.28% of them only received family sex education, and 51.76% of them received both family and school sex education. The CSE comprehensive capacity was significantly different among these four groups. In addition, the SSS score was positively correlated with CSE (including the three indicators: SSS Family, SSS School, and SSS Total).

### 3.2. The Current Sex Education Learning Content among Chinese Junior High School Students

The sex education learning content is shown in Figure 1. A similar tendency between school sex education and family sex education with regard to the proportion of different learning contents was found. The most common content of sex education for both school and family was “mental health during puberty”, and the least involved content for both school and family was “reproduction and contraception”.

### 3.3. The Correlation between Sex Education and CSE Level among Chinese Junior High School Students

As shown in Table 2 and Table 3, junior high school students who received both home and school-based sex education had the highest scores on CSE knowledge, attitude, skills and comprehensive capacity. Those who only received school-based sex education had a significantly higher level of CSE knowledge (12.44 ± 2.89 points) than those who only received home sex education (11.91 ± 3.01 points). In those who have only received family sex education, their CSE attitude level (47.86 ± 5.61 points), skills level (40.71 ± 5.42 points), and comprehensive capacity scores (82.02 ± 9.01 points) were significantly higher than those who have only received school-based sex education.

### 3.4. The Correlation between Subjective Social Status and CSE Comprehensive Capacity

As shown in Table 4, after adjusting for parental accompaniment, the learning experience of sex education, and other covariates, the subjective social status (total) was a positive correlation with CSE comprehensive capacity (β = 0.34, 95 CI% = 0.26, 0.42, *p* < 0.001). After the adjustment for parental accompaniment, the learning experience of sex education, and other covariates, there was no significant correlation between SSS (Family) and CSE comprehensive capacity, and higher standardized coefficients (beta) of SSS were observed for School (beta = 0.15) than those of Family (beta = −0.02).

## 4. Discussion

In this study, the average scores among Chinese junior high school students of CSE comprehensive capacity, knowledge, attitude, and skills were 82.44, 12.43, 47.86, and 40.44, respectively. The CSE comprehensive capacity of Chinese junior high school students was related to both SES and parental accompaniment. Furthermore, after the adjustment for confounders, CSE comprehensive capacity was correlated with the learning experience of sex education and subjective social status. The implementation of school-based sex education in China was 74.23%, which was higher than that of some other Asian countries [17]. The most common teaching content of sex education was “mental health during puberty”, and the least common content is “reproduction and contraception”, which might be related to embarrassment when parents talk about this topic with children [18,19]. However, evidence suggested that sex education including the content of reproduction and contraception can positively influence behaviors that prevent unintended pregnancy and STDs [20,21,22,23]. Hence, a more comprehensive school-based sex education with the teaching of content on reproduction and contraception should be conducted in China to expand understanding of gender equality and reduce school bullying.

CSE teaches not only traditional reproductive health education and promotion, the capacity for promoting health and wellbeing, and an understanding of how the wellbeing of others might be affected by one’s choices, but also an understanding and respect for others’ rights [4,17]. In the present study, we found that family sex education is more positively related to CSE skills compared to school sex education; meanwhile, school sex education is more positively related to CSE knowledge compared to family sex education. These findings could extend our understanding of the learning experience of sex education compared to other sex education-related studies in China [24,25]. The results of the present study suggested that school-based sex education could be more likely to improve the sex education knowledge level, whereas family sex education was more likely to have a positive impact on the related behavior in junior high school students. These results were in line with the studies on the relationship between family environment and adolescents’ cognition and behavior [26,27]. Additionally, these results are in line with the finding that Chinese parents may not have enough sex-education-related literacy to teach their children, therefore school sex education can better promote a higher sex education knowledge level than that within the family [28,29]. As a result, schools could undertake more theoretical knowledge teaching of CSE in the future, whereas family sex education could be more skill-oriented, which could aid in fostering the capacity for decision-making and a better understanding of gender norms and equality, which would be a life-long benefit. Also, health and education professionals shall consider developing more skill-oriented materials for family sex education and more knowledge-oriented materials for school sex education.

Danny Rahal and colleagues’ research on subjective social status suggested that adolescents with a lower SSS may be more likely to have greater sensitivity to threats, resulting in poorer mental health and potentially setting the stage for longer-term mental health problems during adulthood [30,31]. Additionally, CSE is proved to be effective in promoting a better understanding of positive peer relations among adolescents [4]. A significant positive correlation between subjective social status (SSS) and CSE comprehensive capacity was found in this study. Although the mechanism linking SSS and CSE is not fully elucidated and we failed to find similar results, it was suggested that CSE might be helpful in promoting adolescents’ SSS by helping them have a better understanding of healthy peer relations; then, this is more likely to help them be more popular in school. Additionally, this is in line with some cases that studied the effectiveness of CSE in promoting young people’s recognition of relations, gender equity, and social justice, which empowers young people to lead safe and productive lives [23,32,33]. Hence, those who have a better capacity of CSE might gain more respect or appreciation from peers, leading to a higher rank of subjective social status which might have a positive impact on their health and wellbeing.

This study has limitations regarding the following aspects: Firstly, convenience sampling was applied in the present study; therefore, the extrapolation of our results was limited. Secondly, the study was conducted through the internet and all information was self-reported, which could lead to bias. Last, but not least, the results of the present study were based on a cross-sectional study design and could not demonstrate causal associations.

## 5. Conclusions

This study found that the CSE comprehensive capacity of Chinese junior high school students was related to SES, parental accompaniment, the learning experience of sex education, and subjective social status. Furthermore, school sex education had the potential to better promote a higher CSE knowledge level compared to family sex education, whereas family sex education was better at promoting CSE skills for students. Subjective social status was positively correlated with CSE comprehensive capacity, with the mechanism left to be discovered. More studies are warranted to build a reliable CSE scale and standard based on the ITGSE and to promote the conduction and surveillance of CSE globally. Furthermore, the study also put up a hypothesis about the mechanism linking CSE with young people’s subjective social status and physiological health.

## Figures and Tables

**Figure 1 children-09-01201-f001:**
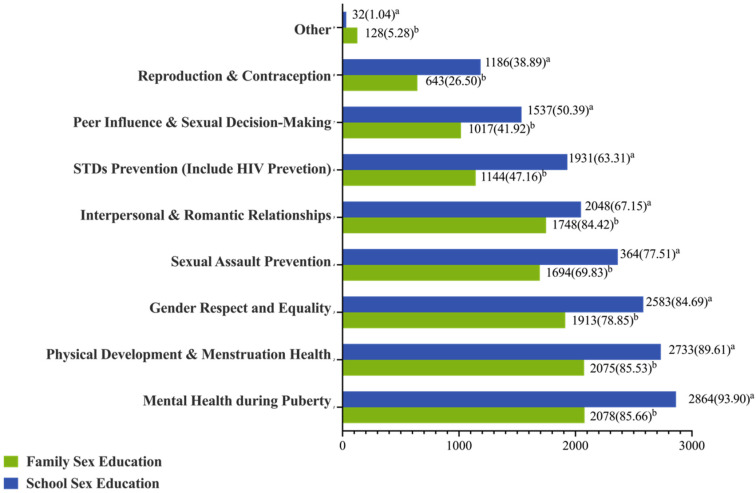
The proportion of the teaching content of sex education in Chinese junior high school students (n (%)). Note: ^a^. the denominator is the number of those who have ever received school sex education, N = 3050; ^b^. the denominator is the number of those who have ever received family sex education, N = 2426.

**Table 1 children-09-01201-t001:** Influencing factors of CSE comprehensive capacity in Chinese junior high school students.

Factors	N (%)/Mean ± SD	Mean ± SD/β (95% CI)	F/t	*p*
Region, % ^a^			23.26	<0.001 ^b^
Yunnan	813 (19.79)	81.08 ± 7.77		
Beijing	793 (19.30)	84.15 ± 8.63		
Sichuan	362 (8.81)	81.19 ± 7.88		
Henan	712 (17.33)	82.15 ± 9.44		
Liaoning	703 (17.11)	81.06 ± 8.59		
Chongqing	726 (17.67)	84.32 ± 8.29		
Gender, % ^a^			3.03	0.082 ^b^
Male	1854 (45.12)	82.18 ± 8.82		
Female	2255 (54.88)	82.65 ± 8.41		
Age, y	0.22	0.01~0.45	2.01	0.045 ^c^
SES	50.01 ± 10.53	0.25 (0.21~0.28)	13.43	<0.001 ^c^
SSS (Family)	5.10 ± 1.90	0.32 (0.18~0.46)	4.55	<0.001 ^c^
SSS (School)	6.10 ± 2.01	0.73 (0.61~0.86)	11.19	<0.001 ^c^
SSS (Total)	11.20 ± 3.27	0.39 (0.31~0.47)	9.54	<0.001 ^c^
Father’s educational background, % ^a^			29.44	<0.001 ^b^
Below elementary school and elementary school	429 (10.44)	80.99 ± 8.00		
Junior high school	1572 (38.26)	81.18 ± 8.45		
Senior high school	907 (22.07)	82.56 ± 8.78		
Bachelor’s degree	1119 (27.23)	84.47 ± 8.44		
Postgraduate and doctoral degrees	82 (2.00)	84.85 ± 8.73		
Mother’s educational background, % ^a^			31.41	<0.001 ^b^
Below elementary school and elementary school	514 (0.92)	81.10 ± 8.63		
Junior high school	1538 (37.43)	81.13 ± 8.26		
Senior high school	873 (21.25)	82.61 ± 8.48		
Bachelor’s degree	1126 (27.40)	84.58 ± 8.62		
Postgraduate and doctoral degrees	58 (1.41)	84.58 ± 9.47		
Sex education learning experience, % ^a^			61.58	<0.001 ^b^
Never learned sex education	760 (18.50)	79.34 ± 8.71		
Only received family sex education	923 (22.46)	81.57 ± 8.37		
Only received school sex education	299 (7.28)	82.02 ± 9.01		
Received family and school sex education	2127 (51.76)	83.98 ± 8.23		
Parental accompaniment, % ^a^			24.91	<0.001 ^b^
Have parental accompaniment	2949 (71.77)	82.85 ± 8.68		
Absence of parental accompaniment	1160 (28.23)	81.37 ± 8.31		
Total	4109 (100)	82.44 ± 8.60		

Note: all the *p*-values are uncorrected. ^a^, values are numbers (percentage, %). ^b^, ANOVA was used. ^c^, regression analysis was used. SES, socioeconomic status. SSS, subjective social status.

**Table 2 children-09-01201-t002:** Correlation analysis of CSE and school/family sex education.

Group	Never Received Sex Education	Only Received School Sex Education	Only Received Family Sex Education	Received School and Family Sex Education	F	*p*
	Mean ± SD	Mean ± SD	Mean ± SD	Mean ± SD		
CSE K	11.26 ± 3.03	12.44 ± 2.89	11.91 ± 3.01	12.91 ± 2.68	67.85	<0.001
CSE A	46.79 ± 5.33	47.47 ± 5.26	47.86 ± 5.61	48.40 ± 5.21	19.66	<0.001
CSE S	39.14 ± 5.45	39.57 ± 4.77	40.71 ± 5.42	41.24 ± 5.06	42.80	<0.001
CSE C	79.34 ± 8.72	81.57 ± 8.37	82.02 ± 9.01	83.98 ± 8.23	61.58	<0.001

Note: CSE K, comprehensive sexuality education knowledge. CSE A, comprehensive sexuality education attitude. CSE S, comprehensive sexuality education skills. CSE C, comprehensive sexuality education comprehensive capacity.

**Table 3 children-09-01201-t003:** Pairwise comparison analysis of school sex education and family sex education on CSE knowledge, attitude, skills, and comprehensive capacity.

		CSE K			CSE A			CSE S			CSE C		
		G1	G2	G3	G1	G2	G3	G1	G2	G3	G1	G2	G3
CSE K	G2	1.18 *											
	G3	0.60 *	−0.53 *										
	G4	1.66 *	0.47 *	1.00 *									
CSE A	G2				0.67								
	G3				1.06 *	0.39							
	G4				1.61 *	0.94 *	0.55						
CSE S	G2							0.43					
	G3							1.57 *	1.14 *				
	G4							2.09 *	1.66 *	0.52			
CSE C	G2										2.22 *		
	G3										2.67 *	0.45	
	G4										4.63 *	2.41 *	1.96 *

Note: *, *p* < 0.05. CSE K, comprehensive sexuality education knowledge. CSE A, comprehensive sexuality education attitude. CSE S, comprehensive sexuality education skill. CSE C, comprehensive sexuality education comprehensive capacity. G1 = never received sex education; G2 = only received school-based sex education; G3 = only received home sex education; G4 = received school and family sex education.

**Table 4 children-09-01201-t004:** Multivariable regression models of the correlation between CSE comprehensive capacity and subjective social status of junior high school students.

Factors	Crude Model *			Adjusted Model **		
	Β (95 CI%)	Beta	*p*	Β (95 CI%)	Beta	*p*
SSS (Total) ^a^	0.34 (0.26, 0.42)	0.13	<0.001	0.28 (0.20, 0.36)	0.11	<0.001
SSS (Family) ^b^	−0.04 (−0.19, 0.11)	−0.01	0.588	−0.10 (−0.24, 0.05)	−0.02	0.191
SSS (School) ^b^	0.70 (0.56, 0.83)	0.16	<0.001	0.62 (0.49, 0.76)	0.15	<0.001

Note: ^a^, SSS (Total) and other confounders analyzed with a multivariable regression model. ^b^, SSS (Family), SSS (School), and other confounders analyzed with a multivariable regression model. * Adjusted for age, region, and gender using the multivariable regression model. ** Further adjusted for parental accompaniment and learning experience of sex education using the multivariable regression model. SSS, subjective social status.

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
