# Peer review of "The Influencing Factors of Comprehensive Sexuality Education Capacity and Its Correlation with Subjective Social Status among Chinese Junior High School Students"

_children, 2022, doi:10.3390/children9081201_

Round 1

Reviewer 1 Report

This is an interesting study that evaluates the factors that influence the capacity of comprenhensive sexuality education and its correlation with subjective social status among Chinese high school students, which, according to the methodological design used, presents some limitations that reduce its validity external.

It is important to clarify the following observations

Title

According to the selection of participating provinces, schools and students can ensure that the study is representative of Chinese high school students.

Methods

What was the criteria for the selection of the six provinces?

The provinces are the same according to sociodemographic and cultural characteristics or they are different

Study limitation: non-probabilistic sampling

Probabilistic sampling was carried out for the choice of regions and schools; not having carried out probabilistic sampling reduces the external validity of the study

You verified if the 26.32% of the participants who did not respond adequately could have modified their results

Why was the sample size calculation not performed?

The subjective social status scale is validated in the population of high school students in the six regions evaluated.

Results: As it was shown that the comprensive capacity of comprehensive sexuality education correspondend to that of the parents and not to that of the high school student when their parents accompanied them. 

Reviewer 2 Report

The paper is clear and well written, and presents an interesting research on the sexual education in Chinese adolescents. While more work could have been done to extend on the literature and contextualize the findings, the paper reports clearly the results and discusses them in the appropriate terms. I have a few minor notes mostly on tables and the limitations.

Major concerns:

- Section 2.1. Provinces are repeated in Line 65 and 73, they are in different orders and are once called provinces, once regions. I would suggest removing the second mention to them on Line 74 as it forces the reader to switch back and forth to compare the two lines to see if they are consistent.

- Table 1. I think that in the footnote B should be in lower case. Also, does the p referred to the uncorrected p-values or to the Bonferroni's corrected p? I would suggest the authors to specify it either in the column's header or in the footnotes.

- One reference is missing on Line 233.

- Limitations. I think the sentence should read "convenience sampling" and not "convenient". Additionally, in what sense a bias because of the format of the questionnaire is expected? Do the authors expect a positive/negative bias toward one or both of the environment where sexual education is received? I would suggest the authors to expand on this point in order to better contextualize their findings.

Minor comments:

- Figure 1. I would suggest the authors to invert the order of the elements in the legend in order to match the order in the chart.

- Some sentences could be expanded for clarity and context. For example, on Line 221 it may be nice to have at list a mention of other Asian countries with their percentages, or other findings for other studies on Line 236. However, freedom should be given to the authors on their writing, so take this just as a suggestion, not a real concern for the article.

- Please verify the meta information at the end of the manuscript, such as the supplementary material and data availability statement.

Overall the paper is of high quality. I would suggest the editor to accept the paper for publications after the minor errors are fixed (e.g. missing reference) and a few clarifications are made (e.g. uncorrected vs corrected p).

Reviewer 3 Report

Though the method being used in this study is quite rigorous and has a large sample size, I feel quite lost while reading the paper, especially while trying to understand the logical relationship between the two major variables, i.e., Comprehensive Sexuality Education capacity & Subjective Social Status (SSS). I am unable to see how this paper had hypothesized the relationship between this two variables prior to the study. It reads like a paper trying to justify the association of two remote/unrelated concepts after conducting the data analysis. Crucial literature review regarding the hypothesis of the aforementioned relationship is totally missing. Discussion part is particularly weak and unsound. It also does not sound appropriate to ask junior high school students about their Subjective Social Status (SSS). Such act might result in negative consequences. I am unable to support this paper for publication. Besides, there are a lot of typing errors and grammatical mistakes, e.g., (page 7) – “Error! Bookmark not defined..”

Round 2

Reviewer 3 Report

I do not feel like the authors have addressed to my concerns directly. There are fundamental problems regarding the theoretical base of this study. Sorry that I am unable to support the publication of this paper in its current form. Anyway, I will respect the editor's decision and thank you for the opportunities to review this paper.

Author Response

Anyway, thank you for your suggestion.